# Molecular Regulation in Dopaminergic Neuron Development. Cues to Unveil Molecular Pathogenesis and Pharmacological Targets of Neurodegeneration

**DOI:** 10.3390/ijms21113995

**Published:** 2020-06-03

**Authors:** Floriana Volpicelli, Carla Perrone-Capano, Gian Carlo Bellenchi, Luca Colucci-D’Amato, Umberto di Porzio

**Affiliations:** 1Department of Pharmacy, University of Naples Federico II, 80131 Naples, Italy; floriana.volpicelli@unina.it (F.V.); perrone@unina.it (C.P.C.); 2Institute of Genetics and Biophysics “Adriano Buzzati Traverso”, CNR, 80131 Rome, Italy; bellenchi@igb.cnr.it (G.C.B.); diporzioumberto@gmail.com (U.d.P.); 3Department of Systems Medicine, University of Rome Tor Vergata, 00133 Rome, Italy; 4Department of Environmental, Biological and Pharmaceutical Sciences and Technologies, University of Campania “Luigi Vanvitelli”, 81100 Caserta, Italy

**Keywords:** dopamine, microRNAs, midbrain, morphogens, Parkinson’s disease, transcription factors

## Abstract

The relatively few dopaminergic neurons in the mammalian brain are mostly located in the midbrain and regulate many important neural functions, including motor integration, cognition, emotive behaviors and reward. Therefore, alteration of their function or degeneration leads to severe neurological and neuropsychiatric diseases. Unraveling the mechanisms of midbrain dopaminergic (mDA) phenotype induction and maturation and elucidating the role of the gene network involved in the development and maintenance of these neurons is of pivotal importance to rescue or substitute these cells in order to restore dopaminergic functions. Recently, in addition to morphogens and transcription factors, microRNAs have been identified as critical players to confer mDA identity. The elucidation of the gene network involved in mDA neuron development and function will be crucial to identify early changes of mDA neurons that occur in pre-symptomatic pathological conditions, such as Parkinson’s disease. In addition, it can help to identify targets for new therapies and for cell reprogramming into mDA neurons. In this essay, we review the cascade of transcriptional and posttranscriptional regulation that confers mDA identity and regulates their functions. Additionally, we highlight certain mechanisms that offer important clues to unveil molecular pathogenesis of mDA neuron dysfunction and potential pharmacological targets for the treatment of mDA neuron dysfunction.

## 1. Overview of the Midbrain Dopaminergic System

In mammals, the dopaminergic neurons are predominantly located in the midbrain. They play an important role in several neural functions including regulation of voluntary movement, attention, motivational and cognitive processing, emotive behavior and reward and responses to unusual or unpleasant experiences. mDA neurons also play a crucial role in decision-making and in reward prediction error signaling. These higher cognitive functions guide complex learning mechanisms subserved by basal ganglia and frontal cortex.

mDA neurons originate in the midbrain floor plate [1,2] and migrate in a tangential direction towards their final location [3,4]. They give rise to the DA neurons in the substantia nigra (SN, A9), which projects to the dorsal striatum via the nigrostriatal pathway, and controls voluntary movement. The floor plate precursors also give rise to two other mDA neuron groups, in the ventral tegmental area (A10, VTA) and the retrorubral field (A8), involved in the regulation of emotion and reward through the meso-cortico-limbic system, which innervates the ventral striatum (or nucleus accumbens) and the prefrontal cortex [5].

The SN, besides including the DA cell bodies in the pars compacta (SNc), encompasses a second half called pars reticulata where DA dendrites are laid out, interspersed among GABAergic neurons, which exert an inhibitory activity on mDA neurons in SNc and, indirectly, on their target area (such as thalamus and superior colliculus). The VTA also comprises a heterogeneous mixture of DA (about 65%), GABAergic (30%) and glutamatergic (5%) neurons.

Degeneration of the nigro-striatal mDA pathway is the cause of Parkinson’s disease (PD) [6], while dysfunction of the meso-cortico-limbic pathway is associated with schizophrenia, drug addiction, attention deficit hyperactive disorder (ADHD), depression and chronic pain [7,8,9,10], see Figure 1.

## 2. Features of Midbrain Dopaminergic Neurons

### 2.1. Molecular Characteristics of Dopaminergic Neurons

A dopaminergic neuron is a typical cell that releases the neurotransmitter dopamine (DA). As mentioned above, DA is involved in important homeostatic responses and motor control in humans. This wide range of adaptive behavioral responses is attained by a relatively small number of mDA neurons (about 400,000–600,000 in human) [11], which develop a very extended axonal arborization, sending their terminals for long-distance in many brain areas, with extensive branching and connections, including en passant synapses. The mDA neurons show a remarkable diversity, recognized only recently [12,13,14]. Molecularly, a dopaminergic neuron express: i) a set of genes common to all neurons; ii) a set of genes necessary for DA synthesis and neurotransmission. The DA synthesis and neurotransmission require genes encoding tyrosine hydroxylase (TH) and dopa decarboxylase (Ddc). They are necessary for the production of DA from its precursor L-tyrosine. The vesicular monoamine transporter 2 (VMAT2), also known as Slc18a2, is necessary for DA packaging into vesicles [15]. Since these genes are expressed in other catecholaminergic neurons, such as noradrenergic neurons, which requires dopamine ß-hydroxylase (Dbh) to synthesize noradrenaline from dopamine, the presence of TH and absence of Dbh characterizes DA neurons. Besides the Central Nervous System (CNS) noradrenergic neurons, the DA synthesis pathway is present also in peripheral blood lymphocytes, in catecholaminergic neurons of the Enteric Nervous System (ENS) and in the gut microbiota [16].

DA released at the dopaminergic nerve terminals activates postsynaptic dopamine receptor D1- or D2-type G-coupled receptors, and presynaptic autoreceptors of the D2-type. The latter are involved in the regulation of DA synthesis, metabolism, and release. The dopamine transporter plasma membrane glycoprotein (DAT), also known as Slc6a3, a member of the sodium/chloride-dependent neurotransmitter transporter family, terminates DA neurotransmission by high-affinity uptake into the presynaptic DA fibers [17]. DAT expression, together with TH, identifies bona fide DA neurons [18].

An additional feature of mDA neurons is the expression of developmental genes such as forkhead box A 1/2 (Foxa1/2), LIM homeobox transcription factor 1 alpha/beta (Lmx1a/b) and Nurr1, also known as Nuclear Receptor Subfamily 4 Group A Member 2 (NR4A2) [19,20]. These genes are not concertedly expressed in DA neurons located in hypothalamus and olfactory bulbs, which do not originate in the floor plate. A number of the above mentioned genes are also expressed in non-DA neurons located in rostral linear nucleus, subthalamic nucleus and ventral premammilary nucleus; therefore, they are not sufficient to identify an mDA phenotype [21,22]. Interestingly, mDA neurons exhibit a pacemaker activity, characterized by spontaneous rhythmic spike activity at low-frequency [23], which persists in vitro in brain slices and in dissociated cultures, representing a hallmark of dopaminergic differentiation and maturation from embryonic or transdifferentiated fibroblasts [24,25].

### 2.2. The Metabolic Rate and Vulnerability of Dopaminergic Neurons

Oxidative stress is a key player in the loss of mDA neurons. mDA neurons have an intrinsic vulnerability, mainly due to their high metabolic demands. In addition, the DA metabolism generates intermediate metabolites that cause oxidative stress [26]; DA degradation generates reactive oxygen species (ROS) and DA oxidation generates many o-quinones, including DA o-quinones, aminochrome and 5,6-indolequinone. Therefore, DA metabolism is important for neuronal redox-homeostasis and viability. Furthermore, the pacemaker activity of DA neurons is accompanied by large oscillations in intracellular Ca^2+^ concentration. The Ca^2+^ in the cytoplasm is relatively free and high, while the levels of Ca^2+^-buffering proteins, such as calbindin, are low. Thus, the Ca^2+^ overload results in mitochondrial oxidant stress, which largely contributes to the degeneration of mDA neurons and the pathogenesis of PD [27]. Recent data have demonstrated that dysfunction in Ca^2+^ signaling may cause the increased nuclear translocation of the transcriptional repressor histone deacetylase 4 (HDAC4) in PD iPSC-derived dopamine neurons and repression of genes that promote neuronal survival [28]. In addition, the misfolding of *α*-synuclein (α-syn, encoded by the *SNCA* gene) causes widespread aggregation of the α-syn protein in the form of Lewy bodies, leading to Lewy pathology, a hallmark of PD [29,30]. α-syn oligomers are highly toxic and their extracellular release activate astrocytes and microglia, generating local inflammation and neuroimmune reaction [31].

Unbiased “omic” approaches have unveiled intracellular and secreted molecular targets involved in the survival, development and cell death of neurons including dopaminergic [32,33,34,35]. In particular, mortalin, also known as Glucose Regulated Protein 75 (GRP75) or mitochondrial heat shock protein 70 (mtHsp70), a protein discovered by means of this methodological approach, has proven to play an important role in the regulation of survival and death of dopaminergic neurons [36]. Mortalin has a multifaceted role since it can interact with a variety of partners within a number of sub-cellular compartments (i.e., mitochondria, cytosol, endoplasmic reticulum and vesicles); nevertheless, in DA neurons, it can regulate their vulnerability. Indeed, mortalin has been found downregulated in the midbrain of PD patients and in preclinical animal models its loss of function leads to mitochondrial proteolytic stress and neuronal death. Interestingly, Parkin and PINK1, two proteins related to PD, are able to rescue the deleterious effects of its loss [37].

Recent evidence has shown that in most PD patients, Lewy pathology is also present in ENS and in the vagus nerve, which express high levels of α-syn. It has been suggested that the vagus nerve could represent a route by which α-syn pathology can bidirectionally spread between the gut and the CNS, thus constituting an important etiological factor in PD [38]. Interestingly, gut dysfunction and gut microbiota modifications occur in most PD patients, supporting that the alteration of the complex cross-talk between brain and gut might play a crucial role in this neurodegeneration [39,40].

### 2.3. ERK Signaling in the Pathophysiology of Dopaminergic Neurons

The aforementioned factors involved in the death of mDA neurons (i.e., ROS, 6-hydroxydopamine (6-OHDA), Ca^2+^ and mitochondrial dysfunction) can regulate, or be regulated by, the extracellular signal-regulated kinases 1/2 (ERK) also known as p42/p44 extracellular mitogen-activated protein kinases (MAPK). ERK is part of a highly conserved signaling module including its direct upstream activator, MEK1/2, a serine-threonine kinase, and RAF-1 or B-RAF, the MEK activators, the latter being highly expressed in the brain. ERK is present within the cytoplasm and translocates into the nucleus upon activation following a variety of stimuli such as growth factors, depolarizing signals and neurotransmitters, including DA, and glutamate (Glu). ERK is also present in the mitochondria of neurons as well as of non-neuronal cells. Alterations of ERK signaling has been involved in the pathophysiology of several neurodegenerative diseases [41,42]. An increase of phosphorylated ERK is found in the cytoplasm and mitochondria of mDA neurons of patients with PD and dementia with Lewy bodies [43]. Moreover, chemical and genetic models of PD and/or neurotoxicity have shown that ERK can exert an essential role in mediating the noxious effects of various pro-cell death stimuli, including the well-known selective dopaminergic toxic compounds such as 6-OHDA, 1-metil 4-fenil 1,2,3,6-tetraidro-piridine (MPTP)/MPP^+^, rotenone and high doses of dopamine. In particular, the study of 6-OHDA has shed light on the role of ERK in DA neurons cell death. Although widely used as an exogenous neurotoxin to generate cellular and animal models of DA neurons cell death, 6-OHDA can be produced in vivo from dopamine and it has been found in the urine of PD patients treated with levodopa [44]. 6-OHDA is capable to generate ROS including H_2_O_2_ and superoxide that are highly toxic in mDA neurons. Interestingly, experimental findings show that the mechanisms of cell death, at least partly, are mediated by the activation of ERK induced by ROS; thus, ERK blockade using MEK inhibitors is able to protect cells from death [45]. Surprisingly, ERK is also a pivotal mediator of many pro-survival as well as differentiation factors such as neurotrophins and growth factors [46]. It has been proposed that the reason underlying such a paradox may lie upon a different kinetic of ERK activation. Indeed, in PC12 cells, as well as in neurons, transient or sustained ERK kinetics determine its noxious or beneficial effects [41,47,48,49,50]. In turn, the different kinetics of activation is shown to mediate, at least partly, the cellular compartmentalization of ERK. Sustained activation of ERK leads to its consistent translocation into the nucleus and cell death, whereas a transient activation, localizing ERK mainly in the cytoplasm, has a protective effect [51]. Cell type and the nature of the stimulus are likely to play an important role in determining the temporal and spatial pattern of ERK signaling. Thus, in order to plan any pharmacological intervention, it appears to be of utmost importance to unveil the cellular and molecular mechanisms underlying pro cell death or survival effects of ERK signaling in mDA neurons. Besides being involved in the pathogenesis of neurodegeneration, ERK signaling also plays an important role in the physiological processes of neurogenesis as well as the maturation of mDA neural progenitors [52]. Neurogenic stem cell niches remain active in the adult brain, giving rise to neurons in the hippocampus and the olfactory bulb, the latter being composed, among the others, of dopaminergic neurons [53]. Notably, in the adult subventricular zone (SVZ), one of the two neurogenic areas, IL-10 is essential to foster neurogenesis by regulating the activation of ERK and STAT 3 in nestin^+^ neural progenitors. Indeed, ERK blockade inhibits neurogenesis in SVZ [54]. Moreover, elegant experiments using D2R−/− mice clearly showed that in midbrain neuronal cultures, Wnt5a controls the number of TH neurons as well as their morphogenic features such as neurite length through ERK activation [52].

## 3. Pathways Involved in the Priming Process of the Ventral Midbrain and mDA Precursor Neurogenesis

Given the role of mDA neurons in PD and neuropsychiatric diseases, intense research activity in the last decades has been directed to unravel the mechanisms of DA phenotype induction and maturation and to elucidate the role of factors involved in the specification, development and maintenance of these neurons and their functions [55]. This remarkable research activity is also based on the assumption that the identification of molecules and cellular interactions involved in the development of the mDA circuits may be the key to generate mDA neurons in vitro for use in regenerative medicine for PD patients.

The development of mDA neurons is a highly coordinated process, which requires the activity of numerous morphogens and transcription factors. The genes encoding this complex network are characterized by time- and cell-specific gene expression patterns that regulate their complex interaction. This articulated differentiation process begins with the establishment of the midbrain-hindbrain boundary by the mutually repressive activities of orthodenticle homeobox 2 (Otx2) in the midbrain and gastrulation brain homeobox 2 (Gbx2) in the hindbrain. In addition, Otx2 suppresses the expression of the transcription factor NK2 Homeobox 2 (Nkx2.2), thus preventing the formation of serotonergic neurons [56]. Therefore, this transcriptional complex is also necessary to suppress non-dopaminergic neural fates. Otx2 also regulates the expression of “proneural” genes (that is, they promote neural differentiation) in the proliferating mDA progenitors, such as Mash1, also known as Ascl1 (Achaete-Scute Family BHLH Transcription Factor 1) and Neurogenin 2 (Ngn2). Transcription factor Foxa2 (also known as HNF3β) is amongst the first expressed in the ventral midbrain. Gain- and loss-of-function experiments indicate that Foxa2 is necessary to produce mDA neurons during development, as well as from embryonic stem cells and fibroblasts in vitro, and to maintain Nurr1 expression in engineered pluripotent stem cells [57]. Foxa2 is also expressed in adult DA neurons and seems to regulate their survival. Mice with one copy of this gene display loss of mDA neurons [58].

### 3.1. The Role of Morphogens and Their Effectors

The commitment of neuroblasts to the mDA phenotype is promoted by the secretion by the floor plate of sonic hedgehog (SHH) [59] and its synergy with fibroblast growth factor 8 (FGF8), secreted in vivo by the isthmus [60]. These two morphogens establish an epigenetic cartesian grid that defines positional information necessary for mDA induction [61]. SHH activates two transmembrane receptors, Patched (PTC) and Smoothened, which elicit an intracellular response with the activation of the zinc-finger transcription factor Gli1 (glioma-associated oncogene 1). The latter therefore represents an early marker of the mDA precursors. Both the floor plate and the isthmus express the basic helix–loop–helix (bHLH) transcription factor Hes1, which suppresses proneural gene expression and induces cell cycle exit, at about day 9 during mouse embryogenesis (E9) [62]. The loss of Hes1 modifies the inductive and repulsive activities of the isthmic organizer. Hes1 regulates the localization and density of mDA neurons [63]. Several genes are required for the correct positioning of the isthmus and the secretion of FGF8. These include the paired box gene 2 (Pax2), Lmx1b, Wingless-type MMTV integration site family member 1 (Wnt1) and Engrailed-1 (En1) (for review, see [64]). It is now established that mDA precursors are floor plate radial glial-like cells [65]. In these precursor neuroblasts, SHH regulates the expression of the Lmx1a, which is necessary to commit midbrain neuroblasts as well as human embryonic stem cells to mDA fate [66,67]. Lmx1a, in turn, activates the expression of Msh homeobox 1 (Msx1), which induces the bHLH proneural gene Ngn2 expression. Lmx1a and Ngn2 gene ablation or silencing result in absence or reduction in the number of mDA neurons. Direct involvement of the morphogen SHH in mDA differentiation has been recently questioned [68]. This conclusion is drawn from the lack of PTC or Gli1 receptors in more mature mDA developing neurons. It has been suggested that other receptors (such as Cdon, Boc and Gas1), in addition to PTC, could modulate SHH signaling activity [69], or that SHH is only indirectly involved in mDA development through early patterning of the midbrain. However, a plethora of in vitro experiments point to a direct role of SHH in mDA differentiation. Thus, in the recent view, SHH is necessary for midbrain progenitors to acquire a dopaminergic cell fate in vitro and in vivo. Furthermore, a direct role of SHH in mDA development is indicated by its induction of the Lmx1a and its downstream gene Msx expression. While Lmx1a expression is maintained in post-mitotic mDA neurons in mice, Msx1 expression is restricted to the mDA neuroblasts [70]. The important role of Msx1 in mDA neurogenesis is revealed by its induction of Ngn2 expression. The latter is also indirectly controlled by Otx2, through the regulation of Lmx1a expression. A well-established role in mDA differentiation in early (mouse E8-9) and more mature (mouse E13) developing mDA neurons has been attributed to the canonical Wnt signaling pathway [71,72]. Indeed, Wnt1 induces the proliferation of mDA precursors by expanding their pool, while Wnt family member 5A (Wnt5a) increases mDA differentiation. These two proteins work by activating the G protein-coupled receptor frizzled, which in turn activates the cytosolic Dishevelled phosphoprotein; the latter inhibits glucose synthetase kinase 3 and blocks phosphorylation of β-catenin and its degradation. β-catenin can thus enter the nucleus, activating a transcriptional response, or causing an increase in intracellular Ca^2+^ through the frizzled receptors and the activation of intracellular protein Dishevelled. β-catenin is a subunit of the cadherin protein complex and the major intracellular signal transducer in the Wnt signaling pathway [73]. Finally, together with Foxa1 and Foxa2, the transcription factors En1 and 2 are also implicated in the determination, differentiation and maintenance of mature mDA neurons and prevent their death from apoptosis [74,75]. Wnt1 and En1 are expressed in graded fashion along the rostrocaudal axis and are more expressed near the mid-hindbrain boundary [76]. Lmx1A, Foxa2 and Otx2 are commonly used to identify mDA progenitors during development [77] and in stem cell cultures [78,79,80]. However, a recent study has revealed that these potential ventral midbrain markers, as well as several other mDA genes, are not sufficient to predict yield or functionality of these cells in vivo since they are also co-expressed in the subthalamic nucleus [22]. Interestingly, current data show that markers expressed by midbrain cells close to the midbrain-hindbrain boundary (i.e., En1, ETS Variant Transcription Factor 5, Canopy FGF Signaling Regulator 1, PAX8 and Sprouty RTK Signaling Antagonist 1) correlate with a successful graft outcome [81].

### 3.2. Genes Involved in the Acquisition and Stabilization of the Functional mDA Phenotype

A number of transcription factors necessary for the development of mDA neurons and their occurrence during development have been described and extensively investigated [82].

Up to E13.5 of mouse embryogenesis, SN and VTA mDA precursors appear indistinguishable, although these two subpopulations of mDA neurons have different markers already at the neural progenitor cell stage. These are the transcription factor Sox6 (or SRY-box 2, sex-determining region Y box 6), known to be involved in establishing pluripotency in stem cells, which is expressed mainly in the SNc neurons, and both Otx2 and the zinc-finger transcription factor Nolz1, which are expressed in VTA precursors [83]. It is worth mentioning that mDA neurons co-express and often co-release GABA/or Glu neurotransmitters. VGLUT2 is more broadly expressed in mDA neurons during development [84,85,86], but its expression pattern is restricted to smaller subsets in adult mouse, in particular in the medial regions of VTA [87,88,89,90,91]. Recent evidence indicates that VGLUT2 co-expression in SNc DA neurons re-emerges in adult mice after DA neuron insult. Interestingly, conditional deletion of VGLUT2 makes DA neurons more vulnerable to neurotoxins, suggesting that the physiological balance of VGLUT2 expression is crucial to DA neuron survival [92].

As previously described, a sequential gene-cohort activation is required to promote the expression of genes, including other transcription factors, that are essential to complete the dopaminergic program in already determined mDA precursors. These genes include Nurr1, which is an “orphan” nuclear receptor belonging to the steroid-thyroid receptor family, the homeogene Pitx3 (Paired-like homeodomain 3), En1 and 2 and Lmx1b. It must be noted that none of these transcription factors alone is sufficient to mature all aspects of the mDA phenotype, indicating that they must act in a coordinated and combinatorial fashion.

Nurr1 (NR4A2) exerts a fundamental role [93]; its expression is under a complex regulation, which includes the activity of a battery of genes as well as depolarization [94]. Nurr1 null mice show loss of mDA neurons, which initially express the dopaminergic transcription factor Pitx3 (see below), but subsequently degenerate and die [95]. Thus, Nurr1 is necessary for the development as well as for the maintenance of mDA neurons. Indeed, Nurr1 regulates the expression of various genes governing DA function, including TH, the rate-limiting enzyme in the biosynthesis of catecholamines, which synthesizes L-DOPA from tyrosine; the aromatic L-amino acid decarboxylase (AADC), which transforms L-DOPA in DA (although its dependence from Nurr1 is not ascertained, as shown in Smits et al. [96]. In addition, Nurr1 regulates the expression of the VMAT2 [97], which concentrates the catecholamines from the cytoplasm into the synaptic vesicles; the cytoplasmic dopamine transporter DAT [98]; the cyclin kinase inhibitor p57/Kip2 [99], which is a negative regulator of cell proliferation; neuropilin-1, a receptor protein of the Vascular Endothelial Growth Factor (VEGF) cytokine family, involved in axon guidance and angiogenesis [97]. Moreover, Nurr1 also regulates the response to neurotrophic factors such as glial derived neurotrophic factor (GDNF) by controlling the expression of its receptors Ret and GDNF Family Receptor alpha (GFRα) [100,101]. Nurr1 also controls the expression of the brain derived neurotrophic factor (BDNF), which in this case could act as an autocrine signal, being involved in a neuroprotective loop for mDA neurons [102]. Nurr1 and Foxa2 act synergistically in microglia decreasing the production and release of proinflammatory cytokines and enhancing the synthesis and secretion of many trophic/pleiotropic molecules (GDNF, BDNF, Neurotrophin 3, SHH, erythropoietin, thioredoxin, transforming growth factor family β and Insulin like Growth Factor 1) [103].

Although olfactory bulb DA neurons express Nurr1, no change in TH in these neurons is observed in Nurr1 null mutant mice, whereas in these mutants there is the loss of mDA. Apparently, Nurr1 plays a role in retina DA neurons differentiation, but has no role in hypothalamic DA neurons [104]. Nevertheless olfactory dysfunction is present in PD patients, and often smell alterations represent a first symptom of the disease [105].

Nurr1 binds to DNA as a monomer, homodimer or heterodimer with retinoid X receptor (RXR) α or RXRγ. In midbrain DA neurons, Nurr1 heterodimerizes with RXRα, and heterodimer activation in mouse and cellular models of PD has a protective effect on mDA neurons, increases TH and other DA functional proteins, augments DA in the striatum and ameliorates and improves symptoms in genetic PD animal models [106]. Nurr1 also has a role in the induction of Pitx3 expression, unveiling a strict relationship between these two transcription factors [107]. Although Pitx3 is expressed in both SN and VTA neurons, the Pitx3-deficient aphakia mouse mutant shows a selective loss of DA neurons only in the SN [108]. Indeed, DA cell physiology does differ between SN and VTA mDA neurons, although these midbrain areas are subject to many common regulatory pathways.

The maturation and survival of mDA neurons in vitro and in vivo is regulated by several families of transcription factors, including BDNF [109,110] and GDNF [111,112]. It has been demonstrated that conditional deletion of GDNF in adult mice causes mDA neuronal loss [113] indicating that GDNF is necessary for mDA neuron survival. Similarly, silencing of gene encoding BDNF in mice results in loss of DA neurons. Moreover, GDNF has been suggested to regulate BDNF expression via a GDNF-Pitx3-BDNF trophic loop [114]. BDNF treatment in animal models before the induction of PD prevented the loss of SN DA neurons and their projections to the striatum. BDNF delivery after induction of PD did not alter the number of DA neurons; however, it elevated the DA level in striatum, increased synaptic plasticity and induced DA axon regrowth [115]. It remains unclear whether the delivery of neurotrophic factors by gene therapy in humans reduces the progression of PD, and the results of the initial trials have been disappointing [115,116]. To date, the delivery of neurotrophic factors to the dorsal striatum of animal models by gene therapy has proved very effective for neurorestoration of dopaminergic neurons [117,118].

Thus, multiple lines of evidence indicate that the network formed by morphogens, transcriptional factors and neurotrophic factors promotes mDA differentiation and survival. In summary, the large number of genes involved in mDA neuron development allows the distinction of five sequential stages schematized in Figure 2.

Recent advances in the field of single-cell transcriptomics and in bioinformatics studies have enabled detection of coordinated gene expression profiles within individual cells [119]. In the last years at least six independent groups have identified molecularly distinct populations of mDA neurons, but their location and molecular signature are only partially overlapping (reviewed by Poulin et al. [14]). For instance, La Manno et al. [120], identified three types of embryonic DA neurons during E11.5 -E18.5 mouse development: i) immature cells expressing TH and other makers of DA neurons, ii) mature neurons expressing TH and DAT and iii) mature neurons expressing—in addition to TH and DAT—also Aldehyde Dehydrogenase 1 family member A1 (Aldh1a1) and LIM Domain Only 3 (Lmo3). Conversely, Hook et al. [121], identified only two embryonic populations at E15 mouse development: a neuroblast population and more mature neurons expressing high levels of TH and DAT. Finally, Tiklova et al. [122] demonstrated that different subgroups of mDA neurons begin to diversify as early as E13 and this process continues during development. These results strongly suggest that after DA neuron maturation and integration into defined neuronal circuits, neurons continue to refine their molecular signature. Additional studies conducted during postnatal development demonstrated that combinatorial expression of Gastrin releasing peptide (Grp) and Neuronal differentiation 6 (Neurod6) defines a dopaminergic subpopulation that resides in the ventromedial VTA and sends projections to the medial shell of the nucleus accumbens [123]. In summary, single-cell gene expression profiling studies, although partially discrepant, highlight the great molecular heterogeneity of the midbrain DA system, challenging the traditional anatomically-based classifications.

How and when DA neuron diversity is generated during development remains unknown, and at the moment, it is not possible to control this subtype diversity during stem cell differentiation. An accurate and molecular definition of mDA subtypes will be crucial for understanding molecular cascades underlying the selective vulnerability in PD, and for enabling an accurate differentiation of iPSC in different DA subtypes.

## 4. Interplay between miRNAs, Dopaminergic Neurons Differentiation, Maintenance and Dysfunction

In recent years, posttranscriptional regulation of gene expression has emerged as an additional important regulatory mechanism in neuronal differentiation, in addition to well defined transcriptional programs. microRNAs (miRNAs) appear crucial in regulating neuronal differentiation and neural circuit development. miRNAs are a class of evolutionary conserved small non-coding single-strand RNA molecules, 18–25 nucleotide long. They are transcribed as long pre-miRNAs, which are processed by subsequent RNAse III activities (Drosha in the nucleus and Dicer in the cytoplasm) forming a duplex, one strand of which is then load in a silencing complex. miRNAs regulate gene expression by binding to mRNAs containing a miRNA recognition sequence [124]. miRNAs are expressed abundantly within the nervous system in a tissue-specific manner [125] and are crucial players in several biological processes including neurogenesis, neuronal maturation, synapses formation, axon guidance, neurite outgrowth and neuronal plasticity [126,127,128]. Alterations in miRNAs contribute to the pathogenic mechanisms in neurodegenerative diseases, including PD [129]. The essential role of miRNAs in neurodegenerative disorders has been highlighted using Dicer knock-out mice [130,131]. Dicer is the cytoplasmatic ribonuclease III and is essential to generate mature miRNA [132]. Several studies have demonstrated that cell type-specific deletion of miRNAs expressed in mDA neurons causes progressive loss of these cells [133], suggesting a pivotal role for miRNAs in mDA neuron formation, survival and function. The relationship between miRNAs and mDA neuron differentiation, as well as miRNAs dysregulation and mDA system dysfunction, such as PD, has been intensively studied in recent years [133,134]. Thus, the detection of miRNAs expressed in mDA neurons could bring about promising information for the early diagnosis and prognosis of PD and offer new potential pharmacological tools.

Overexpression of miRNAs has emerged as a potential additional strategy to improve DA survival in the attempt to counteract the progressive neuron loss in PD and related disorders. In addition, this approach represents an attractive alternative to increase in vitro dopaminergic differentiation starting from undifferentiated ES cells or through the reprogramming of human fibroblasts. In this context, two microRNAs, miR-34b/c and miR-218, have been very promising, since once expressed in combination with specific DA-related transcription factors are able to increase the overall yield of functional DA neurons. The overexpression of miR-34b/c together with the transcription factors Ascl1 and Nurr1 reduces Wnt1 expression and promotes maturation into functional DA neurons that show the typical electrophysiological properties of DA neurons, consisting in a burst of action potential firing pattern [24].

In the same way, miR-218 combined with Neurod1, Ascl1 and Lmx1a is able to promote in vivo the transdifferentiation of astrocytes into fully functional DA neurons and rescues, at least partially, the progressive loss of DA neurons [67]. In both cases, miRNAs, rather than promoting by themselves the differentiation of DA neurons, act to potentiate the effect of specific transcription factors. These results make miRNAs particularly interesting, from a pharmacological point of view, and highlight their potential for therapeutic purposes.

Interestingly, miR-34b/c expression is significantly downregulated in brain areas of PD patients with different degrees of disease, suggesting their role in the early and late phases of the pathology. Here, the miR-34b/c cluster exerts a protective role by regulating Parkin and DJ-1, two proteins associated with familial forms of PD whose expression is affected in PD brain [135]. Similarly, the downregulation of miR-34b/c in differentiated SH-SY5Y neuroblastoma causes cell vitality reduction, affecting mitochondrial function and oxidative stress [135,136].

Among the various miRNAs expressed abundantly in the brain, miR-7a seem to be important to control the site-specific generation of DA neurons in the mouse forebrain [137]. miR-7a inhibits the expression of Pax6, a key transcription factor that controls DA neuron differentiation, along the entire ventricle wall except for the most dorsal part. There, the absence of miR-7a expression allows Pax6 expression, and hence, the occurrence of DA neuron differentiation [137].

In addition, miR-132 has been shown able to regulate the differentiation of DA neurons by directly targeting Nurr1 [134], suggesting its importance for DA neuronal maturation and maintenance. It is now well established that miR-132-3p targets important pathways involved in neuronal plasticity such as long term potentiation, DNA methylation and neuronal cAMP response element-binding protein, N-methyl-D-aspartate receptor and BDNF signaling [138]. On the other hand, the role of miR-132 in PD is not yet totally established. Contrasting evidence has been reported, showing downregulation of miR-132 in the brain of an α-syn transgenic mouse [139], or upregulation of miR-132 in a rat mouse model of PD [140]. In addition, the expression of miR-132-3p was three times higher in the peripheral blood lymphocytes of treated PD patients compared to untreated PD patients [141]. miRNAs also act as protectors of neurons through inhibition of neuroinflammation, apoptosis or autophagy [142]. miR-132 regulates the inflammatory response, while miR-124 plays a protective role in cellular models MPTP-intoxicated SH-SY5Y through the involvement of cell apoptosis and autophagy [143]. At the same time, in MPTP-induced mouse models of PD, miR-124 overexpression promotes DA receptor expression and neuronal proliferation and suppresses neuronal apoptosis by downregulating endothelin 2 through activation of the SHH signaling pathway [144]. Similarly, miR-410 exerts neuroprotective effects in a 6-OHDA cellular model of PD by inhibiting the PTEN/AKT/mTOR signaling pathway [145].

Basically, it can be said that miRNAs play an important role in neurogenesis and survival, by conferring identity and specificity to subtypes of neural cells and modulating the spatial and temporal expression profiles of some fundamental genes. In this context, they represent an important class of molecules useful to better understand PD pathology that could be used as biomarkers in diagnosis and prognosis of this neurodegenerative disorder, as well as new promising pharmacological tools. Knowledge of miRNA-based regulation of the chosen neural phenotype could help develop new strategies for obtaining different types of animal and human neuronal cells to replace damaged mDA neurons.

## 5. Conclusions

The elucidation of the gene network involved in mDA neuron development and function could open new avenues and will be crucial to identify early changes of DA neurons that occur in the pre-symptomatic stages of PD. It can help identify targets for effective new therapies and for regenerative medicine through lineage reprogramming of other cell types (iPSC cells, neurons, glial cells and fibroblasts) into dopaminergic neurons. In addition, understanding the role of genetic and epigenetic factors involved in mDA neurons ontogeny and survival may provide new information for pharmacological manipulation of DA neurons in DA-related disorders, such as schizophrenia, drug addiction, ADHD, depression and chronic pain.

## Figures and Tables

**Figure 1 ijms-21-03995-f001:**
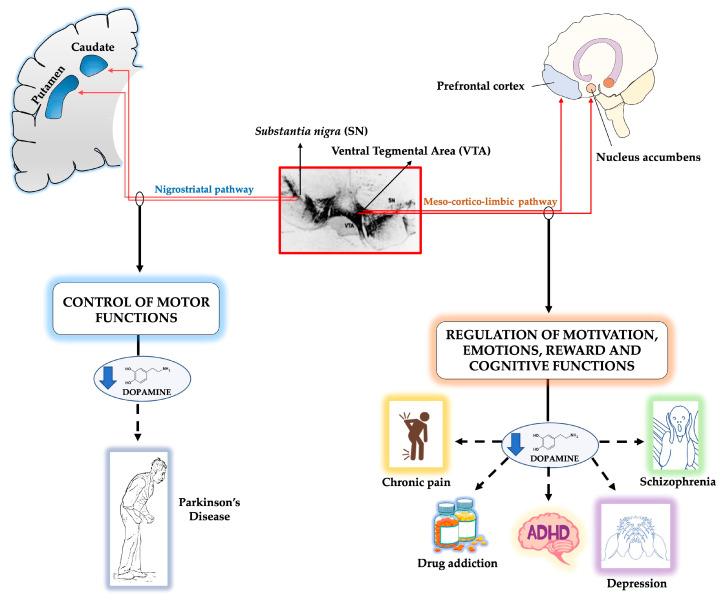
Schematic representation of the midbrain dopaminergic pathways. The figure illustrates in a schematic representation the two dopaminergic pathways originating from the midbrain (delimited by a red rectangle). The ventral tegmental area (VTA) and the substantia nigra (SN) in the adult ventral midbrain are visualized by anti-tyrosine hydroxylase antibodies. On the left: the nigrostriatal pathway connects the SN pars compacta in the midbrain with the dorsal striatum (i.e., caudate nucleus and putamen) in the forebrain. The main function of the nigrostriatal pathway is to control voluntary movement through basal ganglia motor loops. Degeneration of SN dopaminergic neurons leads to diminished concentrations of dopamine in the caudate-putamen. Degeneration of this pathway causes Parkinson’s disease (see text). On the right: the meso-cortico-limbic system pathway connects the VTA in the midbrain to the ventral striatum (i.e., nucleus accumbens) and to the prefrontal cortex. It regulates motivation, emotions, reward and cognitive functions. Dysfunction of this pathway is associated to schizophrenia, drug addiction, attention deficit hyperactive disorder (ADHD), depression and chronic pain (see text).

**Figure 2 ijms-21-03995-f002:**
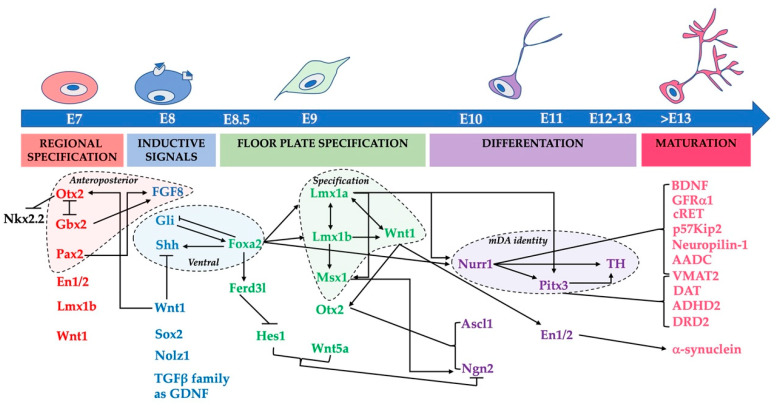
Genetic networks controlling midbrain dopaminergic neuron development in the mouse brain. The diagram summarizes the sequential stages and the molecules involved in mouse midbrain DA neuron embryonic development. The expression of transcription factors (TF) and secreted molecules involved in midbrain regional specification, induction, floor plate specification, differentiation and maturation of the mDA phenotype are indicated, at various embryonic (E) ages. Lines depict possible interactions among these molecules. Arrows indicate stimulatory effects, while perpendicular lines denote inhibitory effects. The orange area, delimited by a dotted line, clusters the TFs and molecules involved in anteroposterior patterning (see text). The light blue area, delimited by a dotted line, clusters the TFs involved in ventral patterning (see text). The green area, delimited by a dotted line, groups the TFs involved in midbrain floor plate specification (see text). The violet area, delimited by a dotted line, collects the TFs involved in mDA identity (see text).

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
