# Peer review of "Molecular Regulation in Dopaminergic Neuron Development. Cues to Unveil Molecular Pathogenesis and Pharmacological Targets of Neurodegeneration"

_ijms, 2020, doi:10.3390/ijms21113995_

Round 1

Reviewer 1 Report

Volpicelli et al. review the molecular determinants of dopamine neuron development in the context of PD pathology and potential therapeutics.

Major Comment

The review touches in several ways on DA neuron heterogeneity, now informed by differential gene expression studies, but does not integrate this new insight.

Other Comments

Title focuses on dopamine neuron development and does not include dopamine neuron pathology although half the review is about pathological mechanisms.

Affiliations include email addresses; corresponding author has two email addresses.

Line 16 and 34: mDA appears to be the abbreviation for midbrain or dopaminergic neurons in the midbrain, but is the abbreviation for midbrain dopaminergic, per Abbreviation listing line 438.

Lines 27 and 29: mDA appears to be the abbreviation for midbrain dopaminergic neuron.

Line 43: should read, “give rise to two other mDA neuron groups, in the ventral tegmental area”.

Lines 57-69: Appears to be the figure caption, but is written as main text.

Line 79: There should be no paragraph break.

Line 102: There should be no paragraph break.

Line 108: Remove the initial “The” from both sentences.

Lines 117-118: Remove, as repetitive: “PD is a neurodegenerative disease 118 that coincides with a dramatic loss of DA neurons within the substantia nigra.”

Line 161: There should be no paragraph break.

Line 170: There should be no paragraph break.

Line 177: There should be no paragraph break.

Line 183: There should be no paragraph break.

Line 196: Colloquial: “a sort of”.

Line 208: Font issue with “floor”.

Line 258: There should be no paragraph break.

Line 288: DA cell physiology does differ between SN and VTA mDA neurons.

Line 300: Reference to Figure 2 should be moved up to the end of line 292.

Lines 248-249: More recent data from Hnasko and Blaes indicate that during embryogenesis all DA neurons express VGLUT2, and thus appear to be capable of glutamate cotransmission.

Line 258: There should be no paragraph break.

Lines 293 – 300: The listing specifies four sequential cell populations, but then the figure has five. The text repeats the listing in the figure in an apparently arbitrary way; it would be better to remove the listing from the text and refer to the figure.

Lines 309-313: Should be part of the caption of Figure 2, but appears to be main text.

Line 324: Do not capitalize “Nervous System”.

Line 330: There should be no paragraph break.

Line 340: Remove “the”.

Line 352: Change “since” to “and”.

Line 344: It is contradictory to state that downregulation in brain areas of PD patients with different degrees of disease suggests a role in the early phases.

Line 374: Remove “the”.

Line 375: Remove “However”.

Lines 383 – 394: The three paragraphs are repetitive and could be consolidated into one.

Line 404 – 406 are repetitive of the previous paragraph and could be consolidated.

Lines 409-410: No quotations needed.

Figure 1: “Schizofrenia” is misspelled.

Figure 2: Boxes start as topics, i.e. Regional Specification” and “Inductive Signals, but then become juxtapositions lacking articles, e.g. “Differentiation mDA neurons” and “Maturation mDA neurons”.

Author Response

Response to Reviewer 1 Comments

Please find below, in red, detailed point-by-point responses to the reviewers’ comments.

Point 1 (Major comment): The review touches in several ways on DA neuron heterogeneity, now informed by differential gene expression studies, but does not integrate this new insight.

Response 1: To address the issue concerning DA neuron heterogeneity the paragraph 3.2 was thoroughly revised and updated with recent papers (lines 455-479).

Other Comments:

Point 2: Title focuses on dopamine neuron development and does not include dopamine neuron pathology although half the review is about pathological mechanisms.

Response 2: The title was modified in: Molecular Regulation in Dopaminergic Neuron Development. Cues to Unveil Molecular Pathogenesis and Pharmacological Targets.

Point 3: Affiliations include email addresses; corresponding author has two email addresses.

Response 3: The corresponding author’s non-institutional email was deleted.

Point 4: Line 16 and 34: mDA appears to be the abbreviation for midbrain or dopaminergic neurons in the midbrain, but is the abbreviation for midbrain dopaminergic, per Abbreviation listing line 438.

Lines 27 and 29: mDA appears to be the abbreviation for midbrain dopaminergic neuron.

Response 4: We deleted the abbreviation from line 35 and we inserted it to line 20. mDA is the abbreviation for “midbrain dopaminergic”.

Point 5: Line 43: should read, “give rise to two other mDA neuron groups, in the ventral tegmental area”.

Response 5: We modified the sentence (line 44).

Point 6: Lines 57-69: Appears to be the figure caption, but is written as main text.

Response 6: We corrected the formatting of the figure legend 1 (line 61 to 72).

Point 7: Line 79: There should be no paragraph break.

Line 102: There should be no paragraph break.

Line 161: There should be no paragraph break.

Line 170: There should be no paragraph break.

Line 177: There should be no paragraph break.

Line 183: There should be no paragraph break.

Line 258: There should be no paragraph break.

Line 258: There should be no paragraph break.

Line 330: There should be no paragraph break.

Response 7: We deleted the paragraph breaks.

Point 8: Line 108: Remove the initial “The” from both sentences.

Response 8: We deleted “the” from both sentences (line 112).

Point 9: Lines 117-118: Remove, as repetitive: “PD is a neurodegenerative disease (118) that coincides with a dramatic loss of DA neurons within the substantia nigra.”

Response 9: We removed the sentence.

Point 10: Line 196: Colloquial: “a sort of”.

Response 10: We changed the sentence (line 242).

Point 11: Line 208: Font issue with “floor”.

Response 11: We modified the font mismatch (line 267).

Point 12: Line 288: DA cell physiology does differ between SN and VTA mDA neurons.

Response 12: We changed the sentence (line 375 to 376).

Point 13: Line 300: Reference to Figure 2 should be moved up to the end of line 292.

Response 13: We moved reference to figure 2 at the end of sentence (line 393).

Point 14: Lines 248-249: More recent data from Hnasko and Blaes indicate that during embryogenesis all DA neurons express VGLUT2, and thus appear to be capable of glutamate cotransmission.

Response 14: To discuss this topic we added a few sentences in paragraph 3.2 (line 316 to 321) and we included additional data, including those by Hnasko and Blaess.

Point 15: Lines 293 – 300: The listing specifies four sequential cell populations, but then the figure has five. The text repeats the listing in the figure in an apparently arbitrary way; it would be better to remove the listing from the text and refer to the figure.

Response 15: We removed the repetitive text.

Point 16: Lines 309-313: Should be part of the caption of Figure 2, but appears to be main text.

Response 16: We modified the formatting of the figure legend 2 (line 396 to 406).

Point 17: Line 324: Do not capitalize “Nervous System”.

Response 17: We removed capital letters (line 490).

Point 18: Line 340: Remove “the”.

                Line 374: Remove “the”.

Response 18: We removed “the” in line 505 and line 540

Point 19: Line 352: Change “since” to “and”.

Response 19: We made the correction (line 520).

Point 20: Line 344: It is contradictory to state that downregulation in brain areas of PD patients with different degrees of disease suggests a role in the early phases.

Response 20: We revised the sentence (line 522).

Point 21: Line 375: Remove “However”.

Response 21: We removed “However” (line 542).

Point 22: Lines 383 – 394: The three paragraphs are repetitive and could be consolidated into one.

Line 404 – 406 are repetitive of the previous paragraph and could be consolidated.

Response 22: We rearranged the three paragraphs to make them more fluent and not repetitive (line 550 to 557).

Point 23: Lines 409-410: No quotations needed.

Response 23: We deleted the quotations (line 587).

Point 24: Figure 1: “Schizofrenia” is misspelled.

Response 24: In figure 1 we modified the misspelled word in “Schizophrenia”.

Point 25: Figure 2: Boxes start as topics, i.e. Regional Specification” and “Inductive Signals, but then become juxtapositions lacking articles, e.g. “Differentiation mDA neurons” and “Maturation mDA neurons”.

Response 25: We modified the boxes in figure 2, respectively in differentiation and maturation.

We would like to thank the reviewer for his constructive criticisms and insightful comments that helped us to improve our review.

Reviewer 2 Report

This review is providing informative collections on the molecular mechanism of dopaminergic neuron’s generation. Its systematic and well-organized format meet criteria being ready for public consumption. There are some minor comments for further improvements:

  1. The layout can be modified more logically. Section 2 is most about molecular and metabolic features of DA neurons. They can be separated to 2.1 and 2.2. The ERK signaling can moved to section 3, which is about the pathway or cell signaling involved in neurogenesis and maturation.
  2. The boost of single cell technologies already changes the understanding of cell identities and their underlying working mechanism on cell development. Lots of papers and database have resolved these detailed neuron types or subtypes. They’re better to be mentioned as in the section 3. Some beautiful works/review are listed below.
  3. Regarding to the genes and effectors part, author missed the other two important factors like BDNF and GDNF, instead putting more emphasizes on Nurr1, which is a bit of lagging back in the current trend.
  4. Few typos/grammatical errors can be found, e.g. Line 19, “Unravel” and Line 20 “elucidate” should be “Unraveling and elucidating”; Line 208 “floor” front mismatches.

  • Ecker, J. R. et al. The BRAIN Initiative Cell Census Consortium: Lessons Learned toward Generating a Comprehensive Brain Cell Atlas. Neuron (2017).
  • Lang, C. et al. Single-Cell Sequencing of iPSC-Dopamine Neurons Reconstructs Disease Progression and Identifies HDAC4 as a Regulator of Parkinson Cell Phenotypes. Cell Stem Cell (2019).
  • Kirkeby, A. et al. Predictive Markers Guide Differentiation to Improve Graft Outcome in Clinical Translation of hESC-Based Therapy for Parkinson's Disease. Cell Stem Cel (2017).

Author Response

Response to Reviewer 2 Comments

Please find below, in red, detailed point-by-point responses to the reviewers’ comments. 

This review is providing informative collections on the molecular mechanism of dopaminergic neuron’s generation. Its systematic and well-organized format meet criteria being ready for public consumption. There are some minor comments for further improvements:

Point 1: The layout can be modified more logically. Section 2 is most about molecular and metabolic features of DA neurons. They can be separated to 2.1 and 2.2. The ERK signaling can moved to section 3, which is about the pathway or cell signaling involved in neurogenesis and maturation.

Response 1: The layout has been modified. We reorganized Section 2: “Features of midbrain dopaminergic neurons”. Thus, it has been subdivided into 3 paragraphs: 2.1 Molecular characteristics of dopaminergic neurons; 2.2. The metabolic rate and vulnerability of dopaminergic neurons; 2.3. ERK signaling in the pathophysiology of dopaminergic neurons. In particular, information related to the role of ERK in the physiology have been added into paragraph 2.1 (line 135 to 146) and, now ERK signaling is an independent paragraph 2.3.

Point 2: The boost of single cell technologies already changes the understanding of cell identities and their underlying working mechanism on cell development. Lots of papers and database have resolved these detailed neuron types or subtypes. They’re better to be mentioned as in the section 3. Some beautiful works/review are listed below.

  • Ecker, J. R. et al. The BRAIN Initiative Cell Census Consortium: Lessons Learned toward Generating a Comprehensive Brain Cell Atlas. Neuron (2017).
  • Lang, C. et al. Single-Cell Sequencing of iPSC-Dopamine Neurons Reconstructs Disease Progression and Identifies HDAC4 as a Regulator of Parkinson Cell Phenotypes. Cell Stem Cell (2019).
  • Kirkeby, A. et al. Predictive Markers Guide Differentiation to Improve Graft Outcome in Clinical Translation of hESC-Based Therapy for Parkinson's Disease. Cell Stem Cel (2017).

Response 2: To address the issue concerning the recent advances in the field of single-cell transcriptomics and in bioinformatics studies on DA neuron heterogeneity, the paragraph 3.2 was thoroughly revised and updated with recent papers, including the review by Ecker et al. (lines 455-456). Lang’s studies were included in the paragraph 2.2 (line 121 to 130) and Kirkeby’s studies at the end of paragraph 3.1 (line 298 to 304).

Point 3: Regarding to the genes and effectors part, author missed the other two important factors like BDNF and GDNF, instead putting more emphasizes on Nurr1, which is a bit of lagging back in the current trend.

Response 3: We discussed in paragraph 3.2 the role of BDNF and GDNF in maturation and survival of dopaminergic neurons (line 377 to 389).

Point 4: Few typos/grammatical errors can be found, e.g. Line 19, “Unravel” and Line 20 “elucidate” should be “Unraveling and elucidating”; Line 208 “floor” font mismatches.

Response 4: We corrected the grammatical errors and the font mismatches.

We would like to thank the reviewer for his constructive criticisms and insightful comments that helped us to improve our review.

Round 2

Reviewer 1 Report

Lines 2-4: Revised title leaves off the object of Molecular Pathogenesis and of Pharmacological Targets. Molecular Pathogenesis of what? Pharmacological Targets of what?

Lines 29-30: The last sentence of the abstract adds mDA dysfunction, but is not complete. Better would be, “We highlight also certain mechanisms that offer important clues to unveil molecular pathogenesis of mDA neuron dysfunction and potential pharmacological targets for treatment of mDA neuron dysfunction.”

Line 30: “mDA” means “midbrain dopaminergic”, so noun, is missing (added above).

Lines 80, 175, 183, 208, 279, 302, 347, 483: Remove several extraneous paragraph breaks.

Lines 128, 184: Remove: “It is worth mentioning that”.

Lines 231, 317: remove extraneous “the”.

Lines 352-353: Sentence would be better placed ahead of previous sentence (line 350), inserted before, “It remains unclear whether…”

Line 381: Change “allowed to detect” to “enabled detection of”.

Line 490: Change “pre-symptomatic pathological conditions, such as PD” to “the pre-symptomatic stages of PD”.

Line 495: Choice of DA-related disorders “depression and chronic pain” appears arbitrary, and moreover are less important conditions than schizophrenia or addiction, for instance.

Lines 496-498: This final sentence is repetitive of the penultimate sentence and further highlights the arbitrariness of the selection of disorders. In Figure 1, the authors select as mDA disorders: PD, schizophrenia, drug addiction, ADHD and depression, with no mention of chronic pain. It would be better to be consistent throughout.

Author Response

Response to Reviewer 1 Comments

Please find below, in red, detailed point-by-point responses to the reviewers’ comments.

Point 1 Lines 2-4: Revised title leaves off the object of Molecular Pathogenesis and of Pharmacological Targets. Molecular Pathogenesis of what? Pharmacological Targets of what?

Response 1: The title was modified in: Molecular Regulation in Dopaminergic Neuron Development. Cues to Unveil Molecular Pathogenesis and Pharmacological Targets of Neurodegeneration.

Point 2 Lines 29-30: The last sentence of the abstract adds mDA dysfunction, but is not complete. Better would be, “We highlight also certain mechanisms that offer important clues to unveil molecular pathogenesis of mDA neuron dysfunction and potential pharmacological targets for treatment of mDA neuron dysfunction.”

Line 30: “mDA” means “midbrain dopaminergic”, so noun, is missing (added above).

Response 2: We changed the sentence as suggested (line 30 to 31).

Point 3 Lines 80, 175, 183, 208, 279, 302, 347, 483: Remove several extraneous paragraph breaks.

Response 3: We deleted the paragraph breaks.

Point 4 Lines 128, 184: Remove: “It is worth mentioning that”.

Response 3: We removed “It is worth mentioning that” in both sentences (line 144 and 203).

Point 5 Lines 231, 317: remove extraneous “the”.

Response 5: We deleted extraneous “the” from both sentences (line 235 and 324).

Point 6: Lines 352-353: Sentence would be better placed ahead of previous sentence (line 350), inserted before, “It remains unclear whether…”

Response 6: We moved the sentence (line 390 to 394).

Point 7 Line 381: Change “allowed to detect” to “enabled detection of”.

Response 7 We changed the sentence as suggested (line 413).

Point 8 Line 490: Change “pre-symptomatic pathological conditions, such as PD” to “the pre-symptomatic stages of PD”.

Response 8: We changed the sentence as suggested (line 541 to 542).

Point 9 Line 495: Choice of DA-related disorders “depression and chronic pain” appears arbitrary, and moreover are less important conditions than schizophrenia or addiction, for instance.

Response 9: We also included schizophrenia, drug addiction, ADHD (line 547).

Point 10 Lines 496-498: This final sentence is repetitive of the penultimate sentence and further highlights the arbitrariness of the selection of disorders. In Figure 1, the authors select as mDA disorders: PD, schizophrenia, drug addiction, ADHD and depression, with no mention of chronic pain. It would be better to be consistent throughout.

Response 10: We removed the sentence at the end of paragraph 4. We added chronic pain in figure 1, in the legend (line 74), and in text (line 60).

We would like to thank the reviewer for his constructive criticisms and insightful comments that helped us to further improve our review.